# The Effects of Adiponectin on the Behavior of B-Cell Leukemia Cells: Insights from an In Vitro Study

**DOI:** 10.3390/biomedicines11092585

**Published:** 2023-09-21

**Authors:** Marta Mallardo, Giulia Scalia, Maddalena Raia, Aurora Daniele, Ersilia Nigro

**Affiliations:** 1Dipartimento di Scienze e Tecnologie Ambientali, Biologiche, Farmaceutiche, Università della Campania “Luigi Vanvitelli”, Via A. Vivaldi, 81100 Caserta, Italy; marta.mallardo@unicampania.it (M.M.); nigro@ceinge.unina.it (E.N.); 2CEINGE Biotecnologie Avanzate “Franco Salvatore” Scarl, Via G. Salvatore 486, 80145 Napoli, Italy; scalia@ceinge.unina.it (G.S.); raia@ceinge.unina.it (M.R.); 3Dipartimento di Medicina Molecolare e Biotecnologie Mediche, Università degli Studi di Napoli “Federico II”, Via Pansini, 80131 Napoli, Italy

**Keywords:** adiponectin, human lymphoblast cell line, JVM-2, non-Hodgkin’s lymphoma, cancer invasiveness

## Abstract

**Background**: Non-Hodgkin’s lymphoma (NHL), the most frequent hematological neoplasm worldwide, represents a heterogeneous group of malignancies. The etiology of NHL remains to be fully elucidated, but the role of adipose tissue (AT) in immune function via the secretion of adipokines was recently recognized. Among adipokines, adiponectin has garnered attention for its beneficial properties. This study aimed to explore the in vitro effects of AdipoRon, an adiponectin agonist, on JVM-2, a lymphoblast cell line used as a representative disease model. **Methods**: JVM-2 cells were treated with different concentrations of AdipoRon to evaluate its effects on viability (via an MTT test), cell cycle distribution (via an FACS analysis), invasiveness (via a Matrigel assay) and colony-forming ability; protein expression was assessed via a real-time PCR (qPCR) and/or Western blotting (WB). **Results**: We found that the prolonged exposure of JVM-2 cells to AdipoRon led to a reduction in their viability due to a cytostatic effect. Additionally, AdipoRon stimulated both the formation of cell colonies and the expression of E-cadherin. Interestingly, the administration of AdipoRon increased the invasive potential of JVM-2 cells. **Conclusions**: Our findings indicate that adiponectin is involved in the regulation of different cellular processes of JVM-2 cells, supporting its potential association with a pro-tumorigenic phenotype and indicating that it might contribute to the increased aggressiveness and metastatic potential of B lymphoma cells. However, additional studies are required to fully understand the molecular mechanisms of adiponectin’s actions on lymphoblasts and whether it may represent a marker of disease.

## 1. Introduction

Non-Hodgkin’s lymphoma (NHL) is the most frequent hematological malignancy worldwide, accounting for 3% of new cancer diagnoses [1]. NHL represents a heterogeneous group of B and T lymphocyte malignancies at various stages of differentiation, some of which remain difficult to treat [2]. Among others, Mantle Cell Lymphoma (MCL) is a rare, incurable subtype of NHL; it is recognized as one of the most aggressive types of lymphoma and is still considered incurable, with a reported median overall survival of about 5 years [3]. The etiology of NHL remains to be fully elucidated, but mounting evidence indicates that immune dysregulation is among the most important risk factors [4]. Actually, obesity is the most common preventable cause of cancer, overcoming cigarette smoking, and has been associated with the risk of hematological cancer [5]. The influential role of adipose tissue in the regulation of immune function via the secretion of biomolecules, known as adipokines [6], was recently recognized. Among adipokines, adiponectin is the most abundantly produced by AT and exerts beneficial effects on a wide range of biological processes [7]. It circulates as oligomers of different molecular weights: low molecular weights (LMWs), medium molecular weights (MMWs) and high molecular weights (HMWs); the latter are considered the most biologically active form [8]. Although adiponectin has been extensively linked to metabolic effects, it is known that it plays a crucial role in immune responses, both innate and adaptive [9]. Adiponectin acts primarily through two known receptors, AdipoR1 and AdipoR2, which are ubiquitously distributed on all cell types and most human monocytes, substantial numbers of B cells and NK cells and a minor proportion of T cells [10]. Regarding its role in inflammation, adiponectin acts by participating in inflammatory and immune responses through the modulation of immune cell activation and cytokine release [10].

Adiponectin appears to inhibit T cell responsiveness, though less is known of its effects on B cells; for example, adiponectin has been shown to inhibit the generation of B cells, probably by inducing prostaglandin synthesis [11]. An increasing number of epidemiological studies have linked low concentrations of adiponectin to hematologic malignancies, including NHL, but with controversial results [12,13,14].

Many tumor cell lines express adiponectin receptors, indicating that adiponectin exerts direct effects on these cells [15]. Indeed, adiponectin possesses several beneficial properties such as an antitumor activity which is carried out by enhancing the receptor-mediated signaling pathways or by indirectly influencing inflammatory responses and tumor angiogenesis [7,16].

In this scenario, the aim of the present study was to investigate the in vitro effects of AdipoRon, a small-molecule agonist of endogenous adiponectin, on a lymphoblastic cell line (JVM-2) used as a representative model of B leukemia. We analyzed the effects of AdipoRon in terms of cell viability, apoptosis, autophagy and cell cycle arrest. Furthermore, we evaluated the effects of the administration of AdipoRon on two other key tumor capacities: invasiveness and the colony-forming ability of B lymphoblasts.

## 2. Materials and Methods

### 2.1. Cell Culture

A human lymphoblast cell line (JVM-2) was kindly provided from the Culture Cell Lines Facility CEINGE Biotecnologie Avanzate, Napoli, Italy. The cells were cultured in Rosewell Park Memorial Institute 1640 Medium (RPMI 1640), purchased from Thermo Fischer Scientific (Thermo Fischer Scientific, Waltham, MA, USA), supplemented with 10% fetal bovine serum (FBS) (Lonza, Basel, Switzerland), 1% L-glutamine (Sigma-Aldrich, Saint Louis, MO, USA), and 1% penicillin–streptomycin (Thermo Fischer Scientific, Waltham, MA, USA) at 37 °C in a humidified atmosphere of 5% CO_2_. All experiments were performed in RPMI and 5% FBS supplemented with various concentrations of AdipoRon (2.5, 5 and 10 μg/mL). The AdipoRon doses were selected based on previous evidence [7,17,18].

### 2.2. Cell Viability Assays

Cell viability was determined via a (3-(4,5-dimethylthiazol-2-yl)-5-(3-carboxymethoxyphenyl)-2-(4-sulfophenyl)-2H-tetrazolium) (MTS reagent) colorimetric assay, as previously reported [17]. Briefly, JVM-2 (5 × 10^3^) cells were seeded in 96-well plates, incubated overnight in a DMEM and 10% FBS medium and, the day after, treated with increasing concentrations of AdipoRon (2.5, 5, 10 μg/mL) (Biovendor, Heidelberg, Germany). Cells receiving DMSO (0.1%) served as vehicle controls and were equivalent to no treatment. After 24, 48 and 72 h of treatment, the cells were stained with MTS reagent, and their absorbance at an optical density (O.D.) of 490 nm was measured using a microplate reader (Model 550, Ultramar Microplate Reader; Bio-Rad, Hercules, CA, USA).

In addition, trypan blue reagent (Bio-Rad, Hercules, CA, USA) was used to measure cell viability. In brief, JVM-2 (1.5 × 10^4^) cells were seeded in 24-well plates, incubated overnight in DMEM 10% and FBS medium and, the next day, treated with AdipoRon (2.5, 5 and 10 μg/mL). After 24, 48 and 72 h, the cells were collected and stained for 5 min with 0.4% trypan blue according to manufacturer’s instructions and counted using a TC10TM Automated Cell Counter (Bio-Rad, Hercules, CA, USA).

### 2.3. Cell Cycle Analysis

JVM-2 cells were plated and allowed to grow overnight to 70–80% confluency. The cells were treated with increasing concentrations of AdipoRon (2.5, 5 and 10 μg/mL) (Biovendor, Heidelberg, Germany) for 24, 48 and 72 h. The cells were then collected, resuspended in 0.3 mL of pre-cooled 1× PBS followed by 0.7 mL of −20 °C ethanol, and maintained on ice for a minimum of 30 min. Finally, the cells were then pelleted via centrifugation, resuspended in a 1× PBS solution of 2.5 mg/mL propidium iodide (Sigma-Aldrich; St. Louis, MO, USA), RNAse (1 mg/mL) (Sigma-Aldrich; St. Louis, MO, USA) and 0.1% triton X-100 in phosphate-buffered saline and incubated for 1 h before a flow cytometry analysis (BD LSRII; Becton Dickinson, San Diego, CA, USA).

### 2.4. RNA Extraction and Quantitative Real Time-PCR

Total RNA was extracted from the JVM-2 cells by using TRIzol Reagent (Thermo Fischer Scientific, Waltham, MA, USA). The RNA concentration was quantified via fluorescence-based detection, using a Qubit 4 Fluorometer (Thermo Fischer Scientific, Waltham, MA, USA). One microgram of total RNA was subjected to reverse transcription with SuperScript III First-Strand Synthesis SuperMix (Thermo Fischer Scientific, Waltham, MA, USA), according to the manufacturer’s instructions. A gene expression analysis was performed using a C1000 Touch Thermal Cycler (Bio-Rad, Hercules, CA, USA) and iQ SYBR Green Supermix (Bio-Rad, Hercules, CA, USA) with the following thermal cycling parameters: 95 °C for 3 min, followed by 40 cycles of denaturation (95 °C for 10 s), annealing (60 °C for 30 s) and elongation (72 °C for 30 s). GAPDH was used as housekeeping gene; fold changes were calculated via the 2^−ΔΔCt^ method. The primer sequences used for the q-RT-PCR are available upon request.

### 2.5. Western Blot Assay

The total protein content was extracted from the cells using a pre-cooled radioimmunoprecipitation assay (RIPA) buffer (Sigma-Aldrich, St. Louis, MO, USA) containing a protease inhibitor cocktail (Abcam, Cambridge, UK). Later, the samples were spun down at 12,500 rpm for 15 min at 4 °C. The supernatant was recovered, and the protein content was quantified via Bradford’s method (Bio-Rad, Hercules, CA, USA). Successively, the samples were diluted in Laemmli buffer 4× and boiled for 5 min at 95 °C.

Then, 30 to 40 μg of total cellular proteins was loaded into a polyacrylamide gel and separated via SDS-PAGE. Thereafter, the proteins were transferred to PVDF membranes (Pierce Biotechnology, Waltham, MA, USA) and blocked with 5% non-fat milk for 1 h at RT; they were then incubated at 4 °C overnight with the following primary antibodies, according to the manufacturer’s instructions: APAF, Caspase-3, PARP, BCL-2, Ubiquitin, p-62, LC3A-B, E-cadherin, GAPDH (Cell Signaling Technology, Danvers, MA, USA), AdipoR1 and AdipoR2 (Santa Cruz Biotechnology, Dallas, TX, USA). The day after, the membranes were incubated with anti-mouse and anti-rabbit antibodies coupled to horseradish peroxidase (Cell Signaling Technology, Danvers, MA, USA). TBS Tween-1X (Thermo Fischer Scientific, Waltham, MA, USA) was used to wash the membranes three times before and after each incubation procedure. Finally, protein bands were detected via a Chemi Doc XRS (Bio-Rad, Hercules, CA, USA), using ECL detection reagents (Pierce Biotechnology, Waltham, MA, USA). To visualize multiple proteins on the same blot, the blots were cut and/or stripped using a stripping solution (Bio-Rad, Hercules, CA, USA), followed by re-incubation with specific primary antibodies.

### 2.6. Colony Formation Assay

For a colony formation assay, 2 × 10^3^ cells were diluted with methylcellulose (Stemcell Technologies, Vancouver, BC, Canada) mixed in RPMI 1640 (1:5), treated with increasing concentrations of AdipoRon (2.5, 5 and 10 μg/mL), and poured into 6-well plates. Colonies were allowed to grow for 14 days and counted manually, using a light microscope as previously described [19].

### 2.7. Matrigel Matrix Invasion Assay

A Matrigel matrix invasion assay was used to analyze the ability of the JVM-2 cells to invade an extracellular matrix, as previously described [20]. In brief, 1 × 10^5^ cells were resuspended in DMEM and 5% FBS containing AdipoRon (2.5, 5 and 10 μg/mL) and plated into the upper well of filter of 8-μM pore size (Costar, Cambridge, MA, USA) covered with Matrigel Matrix (Biosciences, San Jose, CA, USA). A complete medium containing 10% FBS was added to the bottom chambers and used as the chemoattractant. Then, the cells were allowed to invade for 48 h at 37 °C in a humidified atmosphere containing 5% CO_2_. Non-invading cells were removed via washing with PBS and by using a cotton swab, while the invading cells were fixed using 11% glutaraldehyde (Sigma-Aldrich, St. Louis, MO, USA) for 30 min, colored in a crystal violet solution, eluted and quantified at an O.D. of 550 nm.

Additionally, invading cells were also visualized and manually quantified under the microscope. To this aim, 5 × 10^4^ cells were suspended in a DMEM and 5% FBS solution containing the various doses of AdipoRon and allowed to invade the Matrigel. After 48 h, the invading cells were fixed in glutaraldehyde, washed with PBS and stained with blue-fluorescent 4′,6-diamidino-2-phenylindole dihydrochloride (DAPI) (Sigma-Aldrich, St. Louis, MO, USA) nucleic acid, according to the manufacturer’s protocols. Cell counts (five random fields) were determined using a light microscope. The mean number of invading cells was then normalized to untreated cells (NCs) paired equal to 1.

### 2.8. Statistical Analysis

Data are expressed as the means of replicates ± the standard error of the mean (SEM). GraphPad Prism 6 software (GraphPad Software, San Diego, CA, USA) was used to carry out the analyses. *p* values were determined via Student’s unpaired *t*-test (two-tailed). For multiple comparisons, a one-way or two-way ANOVA followed by Tukey’s multiple comparisons test was used. A *p* value < 0.05 was considered statistically significant.

## 3. Results

### 3.1. Adiponectin Receptors Are Expressed in JVM-2 Cells with Up-Regulation at the mRNA and Protein Levels of AdipoR1 Induced via the Administration of AdipoRon 

We confirmed the presence of AdipoRs in the JVM-2 cells at the mRNA level and the protein level, see Figure 1. Then, the JVM-2 cells were treated with AdipoRon (2.5 and 5 μg/mL) for 48 h, and the expression levels of AdipoR1 and AdipoR2 mRNA were determined via a q-PCR. After 48 h of treatment with AdipoRon, a significant increase in AdipoR1 mRNA, without altering AdipoR2 levels, was evident (Figure 1A,B). Western blotting confirmed both the presence of the AdipoRs as well as the specific up-regulation of the AdipoR1 receptor, also at the protein level (Figure 1C,D).

### 3.2. AdipoRon Reduces JVM-2 Cell Viability, Inducing a Cytostatic Effect without Affecting Apoptotic and Autophagy Pathways

The effects of AdipoRon on the viability of the JVM-2 cells were tested by treating the JVM-2 cells, contained in 5% FBS medium, with different doses of AdipoRon (2.5, 5 and 10 μg/mL) for 24, 48 and 72 h and performing an MTT assay. The results demonstrated that for up to 48 h of incubation, AdipoRon did not affect the viability of the JVM-2 cells except at the highest dose of 10 μg/mL; after 72 h, AdipoRon reduced the viability of the cells at all the doses tested (Figure 2A). The percentage of live cells, calculated using trypan blue, confirms that AdipoRon is not toxic for JVM-2 cells until 48 h of treatment except for the 10 μg/mL dose, and the percentage of live cells is reduced in a dose-dependent manner after 72 h of treatment (Figure 2B).

To investigate the molecular mechanisms underlying the cell toxicity induced by AdipoRon for the longest incubation time (72 h), we used a Western blot to analyze some key proteins involved in the apoptotic and autophagy processes (APAF, Caspase-3, PARP, BCL-2, LC3, Ub, and p62C): AdipoRon did not induce apoptosis nor autophagy in the JVM-2 cells (Figure 2C,D).

Successively, cell cycle progression was analyzed via flow cytometry after 24, 48 and 72 h of exposure with the increasing concentrations of AdipoRon (2.5, 5 and 10 μg/mL). Figure 3 shows that after 24 and 48 h of incubation, AdipoRon (2.5 and 5 μg/mL) did not affect the JVM-2 cell cycle, while the 10 μg/mL dose induced an increase in the percentage of G1 cells equal to 10%. After 72 h, an increasing subG1 population (5%, 14% and 25%, respectively, for doses of 2.5, 5 and 10 μg/mL) was observed. These data indicate that AdipoRon caused a slowdown of cell cycle division in the JVM-2 cells at longer incubation times, promoting a G1 phase increase and S decrease at the same time.

### 3.3. AdipoRon Exposure Promotes Colony Formation and Induces E-Cadherin Protein Expression on JVM-2 Cells

The JVM-2 cells were exposed to AdipoRon (2.5, 5 and 10 μg/mL) for 14 days and were tested for colony formation (Figure 4A,B). The data show the induction of a colony-forming capability in the JVM-2 cells exposed to AdipoRon at doses of 2.5 and 5 μg/mL compared to the untreated cells.

To assess whether this induction was connected to the regulation of adhesion molecules, we analyzed the expression of E-cadherin via a Western blot (Figure 4C,D). After the administration of AdipoRon (2.5 and 5 μg/mL) for 48 h, the expression of the protein E-cadherin increased in a dose-dependent manner.

### 3.4. AdipoRon Increases the Invasive Ability of JVM-2 Cells Compared to Untreated Cells

The invasion ability of the JVM-2 lymphoblast cells in response to AdipoRon treatment was analyzed. JVM-2 cells were treated with increasing concentrations of AdipoRon (2.5, 5 and 10 μg/mL) and seeded in the upper chambers of a transwell insert coated with Matrigel. The invading cells, colored with crystal violet, were photographed and quantized after 24 h (Figure 5A,B, respectively). Interestingly, the results clearly indicate that the administration of AdipoRon increased the invasive ability of the JVM-2 cells compared to the untreated cells, with a remarkable induction at doses of 2.5 and 5 μg/mL.

## 4. Discussion

To our knowledge, only one study has focused on exploring the activity of adiponectin on lymphoblastic cells [21]; therefore, the aim of the present work was to investigate the role of adiponectin and its effects on a B lymphoblastic cell line, the JVM-2 cells, taken as a disease model. The cells were treated with different concentrations of AdipoRon, an adiponectin agonist, for up to 72 h to evaluate cytotoxicity, apoptosis, autophagy and cell cycle distribution. Our data indicate a significant role of adiponectin in the cellular processes of then JMV-2 cells, supporting its potential association with a pro-tumorigenic phenotype and suggesting that this adipokine could contribute to the increased aggressiveness and metastatic potential of B lymphoma cells.

Recently, adiponectin has received renewed attention in oncology research as the data from several articles in the literature highlighted the link between the dysregulation of this adipokine and the risk of obesity-associated cancers, including breast, endometrial, lung and hematologic malignancies (i.e., leukemia, lymphoma and myeloma) [16,22]. Accordingly, the majority of epidemiological studies have emphasized a consistent increase in the risk of hematopoietic cancer among a population with excess body weight [22]. Furthermore, hyper-adiponectinemia has been observed in NHL subjects when compared to healthy controls [12,13,14]. The etiology of NHL is not yet fully known, although the dysregulation of the interplay among various organs and tissues, including adipose tissue, may be implicated [4]. Among adipokines, adiponectin could participate in the pathogenesis and progression of NHL and progression, but the underlying molecular mechanisms remain unclear.

Several studies have reported the modulation of adiponectin receptors in hematologic malignancies [23]. Thus, firstly, we confirmed the presence of both AdipoR1 and AdipoR2 receptors on JVM-2 cells. We found that the expression level of AdipoR1 was enhanced after 48 h of AdipoRon exposure while the expression level of AdipoR2 was unaltered, suggesting a different involvement of adiponectin receptors. Furthermore, the administration of AdipoRon to JVM-2 was not effective at modifying cell viability until after 48 h of treatment except at the highest dose; instead, after 72 h of treatment, it reduces cell viability in a dose-dependent manner. To explore the molecular mechanisms underlying AdipoRon-induced cellular toxicity, we evaluated its effects on apoptosis and autophagy, finding that AdipoRon induces no measurable effects on these two JVM-2 cellular processes.

Subsequently, cell cycle progression was analyzed. Our results show that AdipoRon induces a G1-phase arrest of cell cycle progression at longer incubation times (72 h). The G1-phase arrest of cell cycle progression provides an opportunity for cells to either undergo repair mechanisms or follow the apoptotic pathway [24]. Previously, adiponectin was found to arrest the growth of breast and ovarian cancer cells, accompanied by G0/G1 cell cycle arrest and the induction of apoptosis [25]. Such evidence is partially in accordance with our data indicating an arrest of the cell cycle, although we could not find the induction of apoptosis and/or autophagy. To our knowledge, such an effect has never been reported before and might be peculiar to adiponectin’s action on lymphoblast cells. Indeed, only one study analyzed the in vitro effects of adiponectin on lymphoblastic cells, reporting that adiponectin did not produce any inhibition of cell cycle progression, as well as an increase in cell death while regulating different genes, such as CD22, CDH1, IFNG, LCK, MSH2, SPINT2, implicated in the progression of leukemia [21].

In our cell model, beyond the inhibition of cell cycle progression, we found that adiponectin affected other fundamental proliferation characteristics, i.e., the cells’ colony-forming capacity and invasion ability. The number of colonies was enhanced by the adiponectin treatment at a dose as low as 2.5 μg/mL and up to 5 μg/mL; these data are not in accordance with previous results in other cancer cell models that showed inhibitory effects of adiponectin towards the colony formation of osteoblasts and colon cancer [26,27]. Furthermore, we found that the enhancement in colony formation was strictly related to an increase in the expression of the protein E-cadherin. E-cadherin has traditionally been considered a tumor suppressor in several cancer types, including hematologic malignancies [28,29,30]. According to this, Bao B.X. et al. (2019) reported that the down-regulation of E-cadherin was associated with more severe disease progression in children with acute leukemia [29]. Similarly, Qi F.Q. et al. (2023) showed that decreased expression levels of E-cadherin and its increased methylation level were associated with a poor prognosis in a cohort of children with acute leukemia [30]. Contrary to the conventional view of E-cadherin as a tumor suppressor, recent studies have provided support for its potential role as a tumor promoter in several cancers [28]. In fact, E-cadherin has emerged as a regulator of various proliferative pathways, leading to a strong increase in cancer cell characteristics both in vitro and in vivo [31]. Our results suggest that the up-regulation of E-cadherin may be an essential event associated with the anchorage-independent growth of JVM-2 cells which is opposite to its well-known role as a tumor suppressor. In line with our results, Naimo G.P. et al., 2023, showed that exposure to adiponectin increases E-cadherin at both mRNA and protein levels, promoting tumor growth and progression in ERα-positive breast cancer cells [32]. Moreover, it was shown that the expression of E-cadherin is essential for the anchorage-independent growth of oral squamous cancer cells [33]. These findings highlight the complex role of E-cadherin in regulating the behavior of cancer, suggesting that its function may be strongly related to the type of tumor and the stage of disease. It is also to important consider that our cell model was an immortalized B cell line; thus, further research using primary cells is needed to clarify E-cadherin’s involvement in the development and progression of B leukemia and progression.

Many studies in the literature reported that adiponectin reduces the cancer cell invasiveness of several cancer types [7]. Importantly, beyond cell models, animal studies and a xenograft model of prostate cancer demonstrated the effect of adiponectin and its analog effects in inhibiting tumor growth. Conversely, in our leukemia model, adiponectin enhanced the invasion capacity of the lymphoblast JVM-2 cells at a dose as low as 2.5 μg/mL and up to 5 μg/mL.

Such a discrepancy with the published data regarding the effects of adiponectin on colony formation and invasion ability reflects the unique effect of adiponectin on human lymphoblast cells compared to other types of cancer cells. Generally, most of the published data have shown that adiponectin exerts an inhibitory effect towards cancer cell characteristics such as proliferation, survival and invasiveness [23]. On the contrary, our results indicate that adiponectin exerts a positive regulatory effect on cancer cell characteristics, suggesting that it might sustain lymphoblast aggressiveness. In support of our hypothesis of a specific different action of adiponectin on lymphoblasts, it is worth considering that lower serum levels of adiponectin expression have generally been associated with cancer progression and the stage of malignancy in several cancer types [34]. Conversely, hyperadiponectinemia was reported in both adult and childhood NHL patients [12,13,14].

There were some limitations in this study. First, the effects of the administration of AdipoRon in NHL were studied using immortalized tumor cells as a disease model instead of primary lymphoblasts; secondly, this work lacks data on the concentrations of adiponectin present in patients with NHL compared to those present in healthy subjects. This would be useful to determine the potential association of concentrations of adiponectin with the severity of the disease and/or prognosis and thus in evaluating the implications of the clinical application of this protein in the management of NHL. Studies conducted in animal models of the disease that would further improve our knowledge of the relationship between adiponectin and NHL and its potential as a therapeutic target for NHL are lacking. Finally, the molecular signaling involved in AdipoRon’s actions in JVM-2 cells are yet to be clarified, and it should also be considered that a multitude of pathways have been described downstream of adiponectin’s action [35,36,37].

## 5. Conclusions

In conclusion, our findings strongly support the hypothesis that adiponectin plays a significant role as a functional regulator of various cellular processes in JMV-2 cells, a human B lymphoblastic cell line. The potential physio-pathological relevance of our results is based on their indication of the distinctive effect of adiponectin on JVM-2 cells, promoting some fundamental characteristics of these malignant cells. Additional in vitro studies on primary cells as well as in vivo studies focusing on underlying molecular mechanisms are essential to further elucidate adiponectin’s mechanism of action in NHL.

The precise understanding of adiponectin’s function might have clinical implications for adiponectin in B leukemia and hypothesizing innovative therapeutic strategies targeting AT endocrine functions, i.e., the specific silencing of the ADIPOQ gene. A Phase 1/2a clinical trial aimed at evaluating the safety and efficacy of an adiponectin analog ADP355 is in progress [38]. To our knowledge, however, clinical trials in humans for its anticancer activity are not ongoing. It is plausible to think that once their safety is proven, more clinical trials will take place for adiponectin analogs.

## Figures and Tables

**Figure 1 biomedicines-11-02585-f001:**
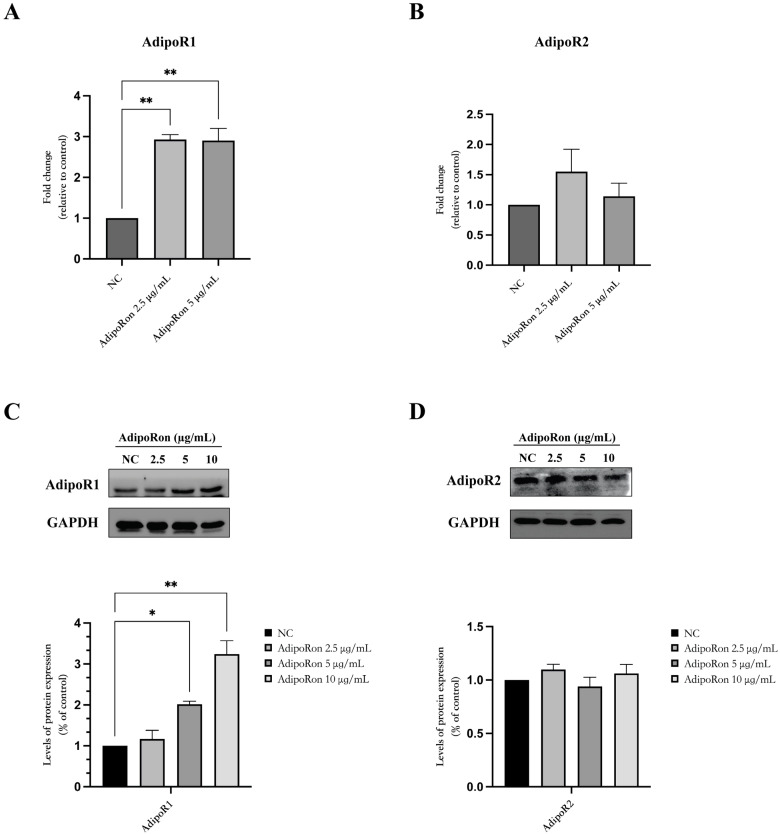
Expression of AdipoRs in the JVM-2 cell line. (**A**,**B**) Expression levels of AdipoR1 and AdipoR2 mRNA were determined at baseline and after 48 h of treatment via a q-PCR; the data were standardized using GAPDH and subsequently quantified via the 2^−ΔΔCt^ method. (**C**,**D**) Western blotting analyses were carried out using GAPDH as internal loading control; a densitometric analysis was performed to normalize the data. Data are reported as the mean ± standard error of the mean (SEM) of two independent experiments performed in triplicate. * *p* < 0.05; ** *p* < 0.01.

**Figure 2 biomedicines-11-02585-f002:**
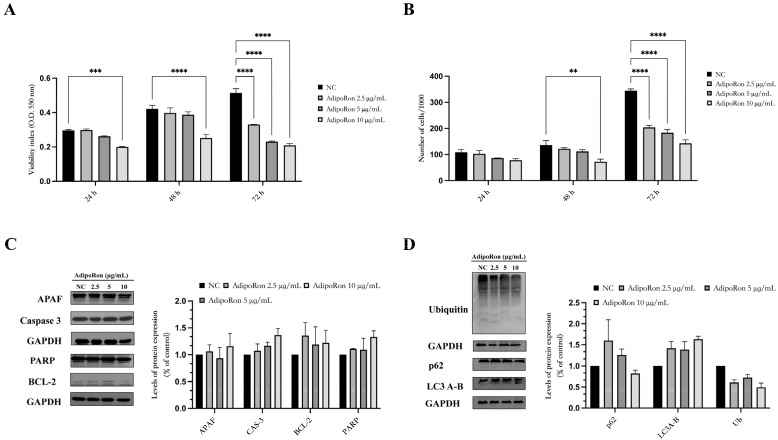
AdipoRon does not affect JVM-2 cell viability, apoptosis or autophagy. (**A**) Cell viability was assessed via an MTT assay after 24, 48, and 72 h of AdipoRon treatment (2.5, 5 and 10 μg/mL). (**B**) The percentage of viability was also calculated using a trypan blue assay after 24, 48 and 72 h. (**C**,**D**) The expression of APAF, Caspase-3, PARP, BCL-2, LC-3, Ub and p62 in JVM-2 cells treated after 48 h of treatment. The relative expression levels of proteins were calculated using GAPDH as an endogenous control. Untreated cells were used as a negative control (NC). Values are expressed as the mean of at least three different experiments ± the standard error of the mean (SEM). ** *p* < 0.01; *** *p* < 0.001; **** *p* < 0.0001 versus NC.

**Figure 3 biomedicines-11-02585-f003:**
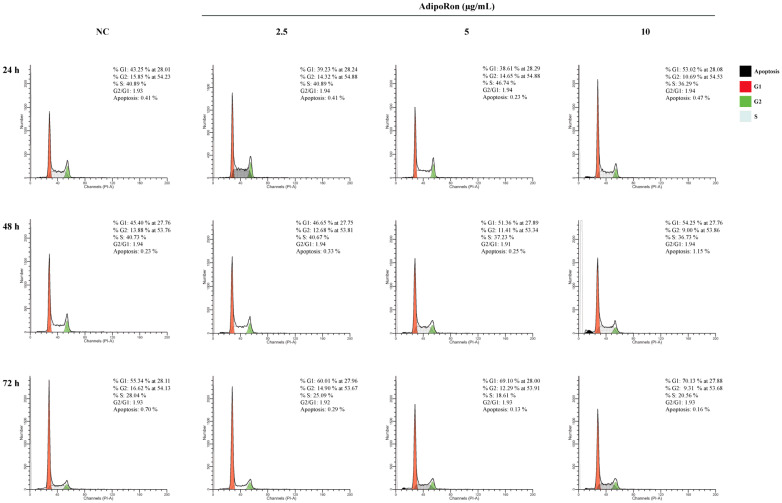
AdipoRon causes a slowdown of cell cycle division in JVM-2 cells at longer incubation times. A cell cycle analysis of JVM-2 cells treated with AdipoRon (2.5, 5 and 10 μg/mL) for 24, 48 and 72 h is shown. Representative FACS histograms of the JVM-2 cells exposed or not (NC) to the AdipoRon treatment are reported.

**Figure 4 biomedicines-11-02585-f004:**
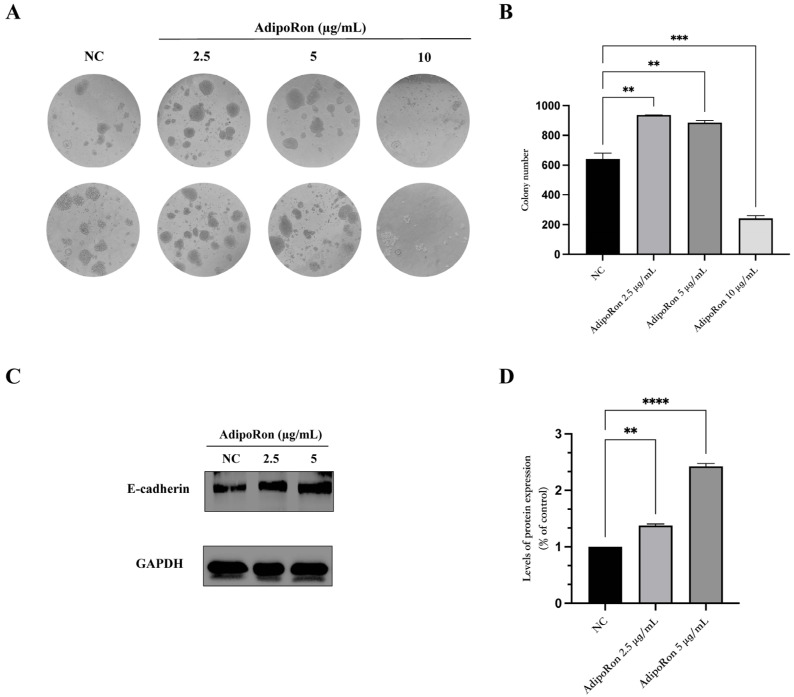
AdipoRon exposure promotes colony formation and induces E-cadherin protein expression on JVM-2 cells. (**A**,**B**) JVM-2 cells were exposed to AdipoRon (2.5, 5 and 10 μg/mL) for 14 days and assessed for colony formation; representative images of colonies and their quantization are reported. The numbers of colonies per well were counted and presented as the mean ± SEM of two independent experiments performed in duplicate. (**C**,**D**) Representative Western blot and pixel quantization of E-cadherin protein after AdipoRon administration (2.5 and 5 μg/mL) for 48 h. GAPDH was used as the internal control. Values are given as mean ± SEM of two independent experiments. ** *p* < 0.01, *** *p* < 0.001 and **** *p* < 0.0001 versus NC.

**Figure 5 biomedicines-11-02585-f005:**
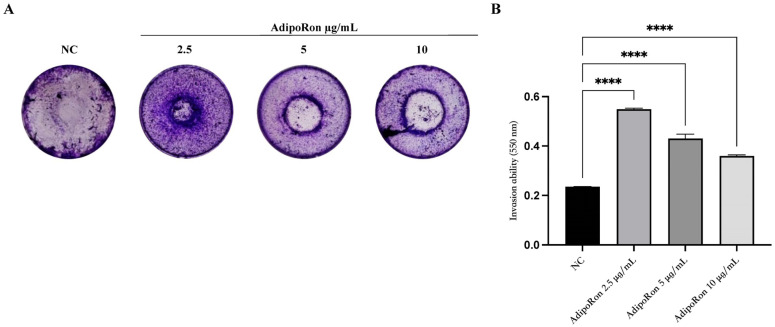
AdipoRon exposure promotes the invasion ability of JVM-2 cells. JVM-2 cells were treated with different AdipoRon doses (2.5, 5 and 10 μg/mL) for 24 h and then plated on a Matrigel matrix and allowed to invade the transwell insert for another 24 h. (**A**) The invading cells were stained and photographed (**B**) and later quantified by measuring the absorbance at an O.D. of 550 nm (**B**). Values are given as mean ± SEM of two independent experiments. **** *p* < 0.0001 versus NC.

## Data Availability

Not applicable.

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
