# Peer review of "The Effects of Adiponectin on the Behavior of B-Cell Leukemia Cells: Insights from an In Vitro Study"

_biomedicines, 2023, doi:10.3390/biomedicines11092585_

Round 1

Reviewer 1 Report

Abstract: contains abbreviations that are not explained.   The introduction is correct, the methodology is presented correctly. The results are presented properly; but figures are technically not good enough.   Figure 1 contains technical errors that require correction; it's about signatures. Figure 3 is very illegible and requires improvement.   This figure contains a color legend, but there are no colors in the engraving. It would be useful to have a summary figure in which the authors would show a possible connection: adiponectin effects on B-cell leukemia cell behavior; what they were able to detect would make it easier to understand the discoveries they made.   There is no information about limitations of the study.  

Minor editing of English language is required.

Author Response

(Reviewer 1)

Abstract: contains abbreviations that are not explained. The introduction is correct, the methodology is presented correctly. The results are presented properly; but figures are technically not good enough.   Figure 1 contains technical errors that require correction; it's about signatures. Figure 3 is very illegible and requires improvement.  This figure contains a color legend, but there are no colors in the engraving. It would be useful to have a summary figure in which the authors would show a possible connection: adiponectin effects on B-cell leukemia cell behavior; what they were able to detect would make it easier to understand the discoveries they made. There is no information about limitations of the study.

Author's Reply to the Review Report (Reviewer 1): We thank the reviewer for the suggestions and modified all figures accordingly. In particular, we have added a summary figure on the effects of adiponectin on the behavior of B-cell leukemia cells. Furthermore, we have added other limitations of the study in the text. Therefore, thanks to the changes made based on reviewers' comments, we hope that the new version of the manuscript and the results are now clearer. 

There are some limitations in this study. First, the effects of AdipoRon administration in NHL were studied using, as a disease model, immortalized tumor cells instead of primary lymphoblasts; further studies on primary cells will be important to confirm the data obtained and delve deeper into the molecular mechanisms underlying the action of adiponectin in NHL. Secondly, this work lacks data on the concentrations of adiponectin present in patients with NHL compared to those present in healthy subjects. This would be useful to determine the potential association of adiponectin concentrations with disease severity and/or prognosis and thus evaluate the implications of the clinical application of this protein in the management of NHL. Finally, studies in animal models of the disease that would further improve knowledge of the relationship between adiponectin and NHL and its potential as a therapeutic target for NHL are lacking.

Reviewer 2 Report

1. The study mentions the effects of AdipoRon on cell viability, cell cycle progression, and E-cadherin expression. However, to provide a more comprehensive understanding, consider discussing the potential molecular mechanisms or signaling pathways involved in these effects. This could involve exploring the literature for existing mechanistic insights or conducting additional experiments to elucidate the underlying pathways.

2. Consider discussing the potential clinical relevance of these findings. How might the observed changes in cell behavior (e.g., increased invasion, altered E-cadherin expression) relate to real-world medical applications or disease states? Providing context for the broader implications of the research can enhance the significance of the findings.

3. The study mentions the use of different AdipoRon doses (2.5, 5, and 10 μg/mL). It would be helpful to provide a rationale for choosing these specific concentrations and to discuss whether these concentrations are relevant to physiological or therapeutic contexts.

4. The effects of AdipoRon were assessed at different time points (24, 48, and 72 hours). Consider discussing the choice of these time points in relation to the expected dynamics of AdipoRon-induced changes in cell behavior. Were these time points selected based on previous literature or preliminary experiments?

5. Since AdipoRon appears to have diverse effects on JVM-2 cells, explore whether there is existing knowledge about the specific mechanisms by which AdipoRon interacts with its receptors and influences downstream signaling pathways. This could aid in interpreting the observed outcomes.

6. Given the observed effects of AdipoRon on JVM-2 cells, discuss the potential clinical applications or therapeutic implications of AdipoRon. Is this compound being explored as a potential treatment for any specific diseases or conditions? If so, elaborate on its potential in clinical settings.

7. If feasible, conduct in vivo experiments using animal models (e.g., xenografts or patient-derived models) to validate the in vitro findings. Assess whether the AdipoRon-induced effects on AdipoR1 expression, cell behavior, and invasiveness are recapitulated in a more complex biological context, providing a stronger basis for potential clinical translation.

These challenging suggestions can significantly advance the understanding of the biological and clinical implications of AdipoRon in the context of JVM-2 cells and beyond.

Moderate editing of English language required

Author Response

  1. The study mentions the effects of AdipoRon on cell viability, cell cycle progression, and E-cadherin expression. However, to provide a more comprehensive understanding, consider discussing the potential molecular mechanisms or signaling pathways involved in these effects. This could involve exploring the literature for existing mechanistic insights or conducting additional experiments to elucidate the underlying pathways.

We thank the reviewer for the observation. We agree that understanding the potential molecular mechanisms or signaling pathways involved in AdipoRon effects on JVM-2 cells would add useful info, however our study represents the first in vitro evaluation of adiponectin action in a model of leukemia.

It is to notice that several signaling pathways have been described downstream adiponectin actions in cancer, in most of the cases related to the cell type. We previously reported in an osteosarcoma and lung in vitro models ERK1/2 activation (Sapio et al 2020; Int J Biochem Cell Biol. 2013). In breast and ovarian cells, it seems to act through AMPK (Mol Cell Biochem. 2019 Nov; 461(1-2): 37–46). The aim of this paper was to clarify at first whether and how an adiponectin analogs act on lymphoblasts. Our study lacks of signaling pathways responsible for AdipoRon actions, but future study will clarify this point.

  1. Consider discussing the potential clinical relevance of these findings. How might the observed changes in cell behavior (e.g., increased invasion, altered E-cadherin expression) relate to real-world medical applications or disease states? Providing context for the broader implications of the research can enhance the significance of the findings.

Reply: To the best of our knowledge, systematic analyzes of the effects of AdipoRon on hematological mechanisms are lacking in the literature. In this study, we used JVM-2 cells, a lymphoblastic cell line, as a preliminary step to study the effects of AdipoRon on the crucial cellular characteristics of these cells and to evaluate its underlying intracellular pathways. The results of this study support the potential association of adiponectin with a pro-tumorigenic phenotype and indicate that it could contribute to increase the aggressiveness and metastatic potential of B lymphoma cells. Our results need to be confirmed in a murine model of NHL disease where the effects induced by AdipoRon on the B cell behavior will be examined in a complex biological context. If confirmed, the next step could be to specifically silence the ADIPOQ gene and verify a possible improvement of the disease in terms of decreased aggression and invasiveness.

  1. The study mentions the use of different AdipoRon doses (2.5, 5, and 10 μg/mL). It would be helpful to provide a rationale for choosing these specific concentrations and to discuss whether these concentrations are relevant to physiological or therapeutic contexts.

Reply: we thank the reviewer for outlining a crucial point. We treated JVM-2 cells with a specific spectrum of AdipoRon concentrations (from 2.5 μg/mL to 10 μg/mL) chosen on the basis of our studies and those of other researchers (Mallardo M. et al, 2023; Nigro E et al., 2021; Akimoto M. et al., 2018). The results obtained in this study demonstrate that the selected doses of AdipoRon do not influence JVM-2 cell viability (except at the highest dose), but induce a series of changes on the main characteristics of these cells. Furthermore, the physiological and therapeutic relevance of these selected doses is supported by studies conducted in preclinical mouse models of various diseases, where concentrations of AdipoRon similar to those we used in this study were utilized. For example, it has been shown that the administration of AdipoRon (5mg/kg/day) inhibits pancreatic tumor growth and tumor proliferation in in vivo mouse model of pancreatic cancer (Messaggio F. et al., 2017). Another study demonstrated that the administration of AdipoRon (1 mg/kg/day) reverses corticosterone-induced depression-like state in mice (Nicolas S et al., 20198). To our knowledge, there is currently no available data in the literature regarding the effects of AdipoRon on in vivo animal models of NHL. Therefore, conducting investigations in preclinical NHL models will provide valuable insights into the potential therapeutic applications of AdipoRon in this disease.

  1. The effects of AdipoRon were assessed at different time points (24, 48, and 72 hours). Consider discussing the choice of these time points in relation to the expected dynamics of AdipoRon-induced changes in cell behavior. Were these time points selected based on previous literature or preliminary experiments?

We treated JVM-2 cells at the three different time points with AdipoRon (24, 48 and 72 h) on the basis of our studies and those of other researchers (Mallardo M. et al, 2023; Nigro E et al., 2021; Diabetologia 2021;64(8):1866-1879. doi: 10.1007/s00125-021-05473-9; Pharmacol Res Perspect 2021;9(6):e00876. doi: 10.1002/prp2.876). The results obtained in the previous studies have shown that AdipoRon is effective already after 24 hours in inducing changes in the main characteristics of cells. On the other hand, time points longer than 72 hours are not usable since cell duplication times do not allow this.

  1. Since AdipoRon appears to have diverse effects on JVM-2 cells, explore whether there is existing knowledge about the specific mechanisms by which AdipoRon interacts with its receptors and influences downstream signaling pathways. This could aid in interpreting the observed outcomes.

Actually, AdipoRon is being largely evaluated both in vitro and in vivo for the treatment of several forms of cancer, including pancreatic adenocarcinoma, myeloma, breast, lung and ovarian cancer. Signaling pathways and molecular mechanisms involved vary according to the cell model and cancer type. Indeed, while AMPK, mTOR and MAPK pathways have been all found downstream AdipoRon incubation, to our knowledge, no data are available about AdipoRon actions on lymphoblasts. Our study represents a first step that outlines a biological effect of adiponectin in this malignancy

  1. Given the observed effects of AdipoRon on JVM-2 cells, discuss the potential clinical applications or therapeutic implications of AdipoRon. Is this compound being explored as a potential treatment for any specific diseases or conditions? If so, elaborate on its potential in clinical settings.

There is a wide literature about AdipoRon use in several metabolic as well as in inflammatory diseases (including several cancer types), both in vitro and in animal models; however, the limitation of this analog use in clinical practice regards the “safety” in non-tumor tissues Akimoto, M. et al. Cell Death Dis. 2018, 9, 804). Aside from AdipoRon, other promising Adiponectin receptor agonists have been discovered over the years. The most promising analog is named ADP355 [Otvos, L. et al. BMC Biotechnol. 2011, 11, 90.]. This peptidomimetic compound has been demonstrated to inhibit cell growth in chronic myeloid leukemia, breast, and prostate cancer [Philp, L.K. et al. Endocr. Relat. Cancer 2020, 27, 711–729; Otvos et al. Front. Chem. 2014, 2, 93.]. ADP355 modulates several intracellular pathways through AdipoR1 binding, including AMPK, mTOR, ERK1/2 and STAT3. In 2019, Allysta Pharmaceuticals initiated a Phase 1/2a clinical trial aimed to evaluate the safety and to explore the efficacy of ADP355 in dry-eye disease (NCT04201574). To our knowledge, however, clinical trials in humans for its anticancer activity are still not ongoing. It is plausible to think that, once the safety of some analogs will be proven, more clinical trials will take place.

  1. If feasible, conduct in vivo experiments using animal models (e.g., xenografts or patient-derived models) to validate the in vitro findings. Assess whether the AdipoRon-induced effects on AdipoR1 expression, cell behavior, and invasiveness are recapitulated in a more complex biological context, providing a stronger basis for potential clinical translation.

These challenging suggestions can significantly advance the understanding of the biological and clinical implications of AdipoRon in the context of JVM-2 cells and beyond.

Reply: We thank the reviewer for the precious suggestion. Exploring the effects of adiponectin and its agonist AdipoRon in a more complex biological context, such as an in vivo model of NHL, might represent the next step towards a deeper understanding of adiponectin's actions in NHL disease. Actually, the administration of AdipoRon has been performed in some in vivo models; i.e. AdipoRon inhibits pancreatic tumor growth and tumor proliferation in in vivo mouse model of pancreatic cancer (Messaggio F. et al., 2017). In another study, it was demonstrated that the administration of AdipoRon reverses corticosterone-induced depression-like state in mice (Nicolas S et al., 20198).

Regarding xenografts, Philp et al. injected the peptide adiponectin receptor agonist ADP355 in mice bearing prostate cancer demonstrating a significant growth inhibition by ADP355, with slowed tumour growth (DOI: https://doi.org/10.1530/ERC-20-0297). Such results are encouraging towards future experiments in patients derived models and clinical trials.

We added this experimental approach in the manuscript as a future plan considering that currently there is no available data in the literature regarding the effects of AdipoRon on in vivo animal models of NHL. Therefore, conducting investigations in preclinical NHL models will provide valuable insights into the potential therapeutic applications of AdipoRon in this disease.

Round 2

Reviewer 2 Report

No comments

Minor editing of English language required